# Latitudinal patterns in the concentrations of biologically utilised elements in the surface ocean

Daisy Pickup<sup>1</sup>, Toby Tyrrell<sup>1</sup> <sup>1</sup>Ocean and Earth Science, University of Southampton, Southampton SO14 3ZH, UK

Correspondence to: ddp1n15@soton.ac.uk

Abstract. Understanding of controls on the spatial distributions of chemical elements in the surface ocean has improved over time. Macronutrients were understood first, followed by dissolved inorganic carbon and alkalinity. Utilising data collected in the Atlantic by the ongoing GEOTRACES programme, controls can now start to be investigated for other elements. Here we investigate the generality of the rule that, in surface waters, higher concentrations occur at higher latitudes. Our analyses of Atlantic GEOTRACES

data show that, after salinity normalisation, all biologically utilised elements except iron follow this rule ( $\rho \ge 0.45$ ). Most elements (nitrate, phosphate, cadmium, barium, and nickel) are even more strongly correlated ( $\rho > 0.6$ ) with latitude. We attribute this pattern to upwelling and/or entrainment of deep water at high latitudes. Although only Atlantic data was analysed here, we predict that this rule will be found to hold true for all oceans in which surface and deep waters exchange more readily at high latitudes. The rule does not hold in the central western Arctic Ocean, where a year-round strong halocline prevents exchange of surface and deep

waters.

#### **1** Introduction

**Macronutrients**: The first distributions to start to be understood were those of the macronutrients nitrate, phosphate and silicate. The main features of their global distributions became apparent as: (1) reliable techniques for measuring concentrations were developed (e.g. Strickland and Parsons, 1972; Gordon et al., 1993), and (2) the first global datasets (of trustworthy data) were

- accumulated (e.g. Brewer et al., 1986; Moore, 1984). One main feature of the global pattern was realised early on that macronutrient concentrations in the open ocean are generally higher towards the poles and lower at low latitudes (Levitus et al., 1993). There is a difference between low and mid latitudes (where, away from upwelling regions and river mouths, concentrations of macronutrients are typically low at all times of year) and high latitudes (where concentrations are either high in all seasons or else are high in winter and low in summer) (Takahashi et al. 1993; Conkright et al. 2000). Although conventional techniques
- struggled previously to measure phosphate accurately at lower concentrations, emerging datasets of nanomolar measurements only serve to reinforce this picture of a general low-latitude vs high-latitude divide (Martiny et al 2019).

The main driver of this general pattern in macronutrients is upwelling. It was established in the 1990's and 2000's that the Southern Ocean and subarctic North Pacific are both iron limited (de Baar et al., 2005; Boyd et al., 2007) and that this is the reason for the

- 30 year-round high levels of residual nitrate, phosphate and silicate in surface waters. Furthermore, the cause of the iron-limitation in these regions is tied to upwelling, because deep ocean water (prior to being upwelled) is more deplete in iron relative to phytoplankton need (more strongly iron-limited than nitrate- or phosphate-limited) (Moore et al., 2016). Therefore, when phytoplankton blooms take place in HNLC (High Nutrient, Low Chlorophyll) regions following upwelling or deep winter mixing, iron runs out first and large amounts of macronutrients are left behind (Moore et al., 2016). The locations of the three main HNLC
- areas, the Southern Ocean, subarctic North Pacific and eastern Equatorial Pacific, all coincide with places where deep water is brought to the surface.

**Alkalinity**: Because of interest in the ocean's role in climate through its uptake of carbon dioxide from the atmosphere, large datasets of two other biogeochemical variables, DIC and alkalinity, have also been accumulated over the last 20 years. The first large global data sets containing alkalinity data (GLODAP and GLODAPv2; Olsen et al., 2016, 2019; Key et al., 2015) have shown

- a more complicated picture for alkalinity than for macronutrients. It is apparent that raw (untransformed) alkalinity data does not show a strong correlation with latitude. Instead it exhibits a more complex distribution in the surface ocean, with highest values in the subtropical gyres, lower values close to the Equator and intermediate values towards the poles (Lee et al., 2006; Millero et al., 1998; Takahashi et al., 2014; Carter et al., 2014). Data analysis reveals a strong correlation with salinity (Millero et al., 1998; Friis et al., 2003; Jiang et al., 2014; Fry et al., 2015); high alkalinity values co-occur with high salinity values in the subtropical gyres
- because both are produced by an excess of evaporation over precipitation. Evaporation (removal of fresh water containing no dissolved ions) raises the concentrations of all the dissolved elemental constituents left behind in seawater and therefore also raises total alkalinity (TA), because TA is a weighted sum of ionic concentrations.
- Although not evident in unprocessed alkalinity data, a hidden high-latitude elevation is revealed when evaporation/precipitation effects are removed (Fry et al., 2015). Moreover, when the effects of other processes known to influence alkalinity are also removed, to leave behind a tracer controlled only by calcium carbonate production and upwelling of dissolution-affected deep waters, it is seen to be upwelling that drives the high latitude elevation in salinity normalised TA (nTA) (Fry et al., 2015). This effect is masked in maps of TA but is apparent in maps of nTA and of related tracers such as potential alkalinity (Takahashi et al., 2014) and Alk\* (Fry et al., 2015).

**Dissolved inorganic carbon (DIC)**: Understanding of the distribution of DIC in surface waters has advanced at the same time as that of TA. A low-latitude to high-latitude gradient is apparent in non-normalised DIC data (Lee et al., 2000; Key et al., 2004; Takahashi et al., 2014) and even more strikingly apparent in salinity-normalised data (nDIC) (Wu et al., 2019). This latitudinal pattern has traditionally (e.g. Follows and Williams, 2011) been attributed to the temperature dependence of  $CO_2$  solubility (cold

water holds more CO<sub>2</sub> gas than warm water; if both are in equilibrium with the same atmospheric CO<sub>2</sub>; in addition, seawater with higher CO<sub>2</sub> also has higher DIC, all else being equal). A recent analysis (Wu et al., 2019) has shown, however, that the latitudinal gradient is driven in more or less equal part by upwelling/entrainment as it is by temperature-induced solubility variation.

Most recently, higher surface nickel concentrations at either end of an Atlantic-long transect have been reported (Middag et al., 2020).

This new understanding of controls on DIC, alkalinity and macronutrients, as well as the recent nickel observations, raise the question as to whether there is a general pattern for all biologically utilised elements. Given that: (1) NO<sub>3</sub>, PO<sub>4</sub>, SiO<sub>4</sub>, DIC, TA and Ni all show tendencies towards elevated values at high latitudes, and (2) in all cases except Ni this has been attributed to deep

water inputs, then it is reasonable to ask whether this is a general pattern that holds true for all biologically utilised constituents in seawater. In this paper we investigate the hypothesis that:

"All bioutilised elements are present at higher concentrations in high latitude than in low latitude surface waters"

where bioutilised elements are defined here to be those which are taken up from surface water by organisms and which, as a result, increase in concentration with depth in the ocean. We distinguish here between bioutilised and biounutilised elements. These 2 categories map onto the 3 categories of an earlier scheme (Broecker and Peng, 1982): our bioutilised category encompasses both biolimiting and biointermediate categories of the earlier scheme; our biounutilised corresponds to their biounlimiting. In this study we do not care whether elements are limiting nutrients for biological production, we care only whether (1) an element is transported

**Biogeosciences** 

- to depth by the particle flux, and (2) this gives rise to a vertical gradient in the concentration of that element. Following the earlier scheme, some essential elements for plankton growth, such as manganese (Raven, 1990), are in this way classified as biounutilised, because their uptake and vertical transport does not result in a vertical gradient of increasing concentration with depth (Fig. 1; see also Sarmiento and Gruber, 2006). Conversely, some elements are classified here as bioutilised even though they are not required by plankton but rather are taken up "by accident" because of their chemical similarity to useful elements (for instance Ge, which
- is taken up inadvertently by diatoms in place of Si (Azam and Volcani, 1981), and increases with depth as a result (Sarmiento and Gruber, 2006)).

Figure 1: Depth profiles for elements in the Atlantic Ocean. Data from GEOTRACES IDP2017v2, cruise GA02, station 33°W 22°S. The prefix 'n' indicates salinity normalisation of concentrations.

Here we use data from the international GEOTRACES programme to explore the extent to which this hypothesis holds true (for instance, as just described, we know it to be true for nTA but not so much for TA). GEOTRACES is the first programme to make

a global survey of the distributions of a large number of different elements, including trace elements and micronutrients as well as the more commonly measured variables. Here we look to see if the rule of high latitude surface enrichment applies also to the wider suite of GEOTRACES variables.

# 95 2 Methods

An earlier global survey campaign (WOCE, the World Ocean Circulation Experiment (Chapman, 1998]) measured macronutrients (NO<sub>3</sub>, PO<sub>4</sub>, SiO<sub>4</sub>) and carbonate chemistry (e.g. DIC & TA) on a global scale. Large data syntheses now exist for macronutrients (WOCE, WOA) and carbonate chemistry (GLODAP) (Olsen et al., 2016, 2019; Key et al., 2015). GEOTRACES is the first programme to measure a larger number of different biogeochemical variables in a comprehensive manner on a global scale. The

- GEOTRACES second Intermediate Data Product (IDP2017v2) contains bottle data from 39 cruises, collected during the period 2007 to 2014. Only dissolved concentration data is used here; particulate concentrations are not included in our analyses. The recommended methods and protocols used to measure the different dissolved element concentrations are included in the GEOTRACES cookbook (available at: https://www.geotraces.org/cookbook). Subsequent to collection, the data was subjected to quality control procedures and, where such procedures indicated systematic offsets, adjustment (Schlitzer et al., 2018). Table 1
- shows which elements were included in our study and on which GEOTRACES cruises they were measured. In this study we only included those GEOTRACES cruises for which final data has been released (the programme is part-way through). At the time of writing, sufficient GEOTRACES data to test the hypothesis of this paper was only available for the Atlantic; for this reason, data from other ocean basins is not considered further here. Figure 2 shows the geographical distribution of the Atlantic data for the various elements.

<sup>110</sup> 

| Table 1: G | EOTRACES | sections and | elements meas | sured on then | i. Only | the cruises and | elements used in | a this studv | are shown her |
|------------|----------|--------------|---------------|---------------|---------|-----------------|------------------|--------------|---------------|
|            |          |              |               |               |         |                 |                  |              |               |

| Section | Dates                                                                                                                       | Elements analysed                                                             |  |
|---------|-----------------------------------------------------------------------------------------------------------------------------|-------------------------------------------------------------------------------|--|
| GA01    | 15/05/2014 - 30/06/2014                                                                                                     | Al, Pb                                                                        |  |
| GA02    | $\frac{28/04/2010 - 26/05/2010}{11/06/2010 - 08/07/2010}\\01/03/2011 - 07/04/2011$                                          | Al, Ba, Cd, DIC, Fe, La, Pb, P, SiO <sub>4</sub> ,15<br>TA, Y, Zn             |  |
| GA03    | $\frac{15-10/2010-04/11/2010}{06/11/2011-11/12/2011}$                                                                       | Al, P, SiO <sub>4</sub>                                                       |  |
| GA04    | $\begin{array}{l} 14/05/2013-05/06/2013\\ 13/07/2013-25/07/2013\\ 25/07/2013-11/08/2013\\ 05/05/2013-01/06/2013\end{array}$ | Al, Cd, La, Y, Zn<br>120                                                      |  |
| GA06    | 07/02/2011 - 19/03/2011                                                                                                     | Al, P, SiO <sub>4</sub>                                                       |  |
| GA10    | 18/10/2010 - 22/11/2010                                                                                                     | Al, Ba, Cd, DIC, Fe, La, Pb, P, SiO <sub>4</sub> , TA, Zn                     |  |
| GAc01   | 16/11/2007 - 13/12/2007                                                                                                     | Fe, La, P, SiO <sub>4</sub> 125                                               |  |
| GIPY04  |                                                                                                                             | Al, Ba, Cd, DIC, Fe, Mn, NO <sub>3</sub> , PO <sub>4</sub> , SiO <sub>4</sub> |  |
| GIPY05  | 10/02/2008 - 16/04/2008                                                                                                     | Al, Ba, Cd, DIC, Fe, P, Si, TA, Zn                                            |  |