# Peer review of "Latitudinal patterns in the concentrations of biologically utilised elements in the surface ocean"

_Biogeosciences, 2020_

## Referee Comment (RC1) · Anonymous Referee #1 · 12 Nov 2020

This manuscript about latitudinal patterns of trace elements is generally well written. It is based on existing data from the GEOTRACES program and aims to test the hypothesis that nutrient type elements occur at higher concentrations at higher latitude, notably the Southern Ocean. The fact that they were able to proof this hypothesis is not at all surprising to me given that nutrient type elements are also referred to as 'accumulated' type elements as they accumulate in older (deep) water. Besides the nutrient type profile (low in surface waters and concentrations that increase with increasing depth) this also leads to a well-known and strong interbasin fractionation where concentrations are higher in the old deep North Pacific or deep Southern Ocean compared to the relatively young deep North Atlantic. As acknowledged in the introduction and discussion of this

ms, upwelling of old deep water in the Southern Ocean thus leads to supply of macro-nutrients. However, this inherently also supplies other nutrient type (trace) elements to surface waters (but not Fe that is subject to scavenging, hence has a hybrid type distribution (Bruland et al., 2014)), and Fe limitation results in 'left-over' nutrients. In the North Atlantic, deep mixing also leads to supply of nutrient type elements to surface waters, albeit lower than compared to the Southern Ocean due to lower deep water concentration in the Atlantic, and seasonal Fe limitation (e.g. Achterberg et al., 2018) results in some 'left-over' nutrients. So while the authors did prove their hypothesis using statistical tests, this hypothesis is actually a well-established concept, not only for the macro nutrients, but also the 'nutrient-type' trace metals (hence their classification as nutrient-type aka as recycled or accumulated type). As far as I can tell, the conclusions of this manuscript are also a main message of any chemical oceanography text book, except for the lines on the Arctic where the authors seemingly missed that the position in the global conveyor (with related absence of old deep water that is strongly enriched in nutrient type elements) is important. Moreover, established concepts regarding the importance of sources, sinks and chemistry of different elements are ignored and I disagree with the notion that recent work did not focus on latitudinal patterns (see specific comments).

Overall, I'm afraid I do not see any novel contribution of this manuscript and therefore cannot recommend it for publication in its current form.

Specific comments

Line 16 distributions of elements in the oceans (there are many distributions that were understood much earlier) Intro Jumps straight into macro nutrient distributions followed by alkalinity without any context or connection between the subsections Line 29 iron and light limited Line 52 awkward sentence. Line 64/65 what is the point of this stand-alone sentence? Similar observations for Cd and Zn by the way Line 69-70 for Ni is was attributed to upwelling of deep water (direct citation: 'The higher concentrations in the Southern Ocean are most likely due to upwelling of older deep water in this region

whereas in contrast, the Arctic is largely supplied by nutrient poor surface water transported north with the Gulf stream' and also depicted in figure 7 of this paper. Similar arguments for Cd and Zn in Middag et al., 2019, 2020) Line 75-80 I find it extremely odd to call the bio-essential element Mn 'biounutilised' whereas it has been shown to limit productivity in the Southern Ocean. Actually, in the Southern Ocean, Mn would be classified as bioutilised (probably bio-utilised is more readable) as concentrations are depleted in the surface and increase with depth (e.g. Middag et al., 2011), whereas Fe in parts of the equatorial Atlantic would be biounutilised (probably bio-unutilised is more readable) as concentrations are elevated in the surface and decrease with depth (e.g. Rijkenberg et al., 2014). Line 89 to what salinity is the data normalized? Line 110 table 1 why is Mn data from GA02 and GIPY 05 ignored? Line 190 why was this based on one individual station (see also previous comment) Line 310 this distribution of Mn and Fe is well known and related to the chemistry of the elements (both subject to oxidative scavenging), biological utilization and notably the presence of strong sources at low latitude (mainly Saharan dust deposition at low latitude, but also fluvial input and reducing sediments) whereas these sources are lacking or much reduced at higher latitudes. The fact that Mn and Fe are low in the HNLC Southern Ocean is something that can be found in any text book or review paper on chemical oceanography addressing these elements. Line 312 For Al and Pb this distribution (e.g. Bridgestock et al., 2016; Middag et al., 2015) is well known and again related to sources and sinks in its biogeochemical cycling. Section 4.3. This is basically a brief summary of a text book on chemical oceanography (As a matter of fact, one of the re-occurring questions I ask in the exam about my chemical oceanography lectures is to explain the higher concentrations of nutrient type elements in the higher latitude regions, notably the Southern Ocean) Line 355 excess of evaporation over precipitation should be accounted for in the salinity normalization. Line 356-357 similar for Mn; and presence and absence of sources such as atmospheric dust, (reducing) sediment, fluvial input, anthropogenic sources etc. 4.5 Deep waters in Arctic Ocean are also not particularly enriched in nutrient type elements like the Southern Ocean as deep waters here are much younger,

i.e. Arctic Ocean sits mainly at the beginning of the ocean conveyor with inflow of nutrient poor Atlantic surface water and only modest amounts of old pacific deep water (see large body of GEOTRACES work in Arctic from both during IPY as well as recent expeditions) Line 375-3-77 I have to disagree here, a main point of those recent papers was the importance of high nutrient (incl nutrient type trace metals) high latitude waters and their influence on both the horizontal (meridional) and vertical distributions (and coupling between elements) at lower latitudes. Line 389 this was a main conclusion of many recent papers (e.g. Middag et al., 2019; Middag et al., 2020; Middag et al., 2018; Roshan et al., 2018; Roshan and Wu, 2015a; Roshan and Wu, 2015b; Vance et al., 2017; Weber et al., 2018) and the lack of Fe supply relative to macro nutrients in upwelling regions is about as old as the term 'HNLC'. 4.7 point 2; given the absence of a strong dust source over large parts of the Pacific, there will be differences for some elements (e.g. the high concentrations of Al, Fe and Mn at low latitude are not found). Moreover, part of the equatorial Pacific is an HNLC region with elevated concentrations of nutrient type elements point 3; this is well known, hence the high-latitude North Pacific is a HNLC region whereas the high latitude North Atlantic only has minor inventories of 'left-over' macro nutrients at the end of the phytoplankton growth season and only experiences seasonal Fe limitation (end of season). point 4: except those with a strong fluvial influence, see recent work on metals in the Arctic trans polar drift. Also noted in recent work on global or Atlantic distribution of Cd, Zn an Ni. Conclusions The statement 'presumably because of its role as the limiting nutrient for primary production in upwelling regions' does not explain anything; the limiting nutrient is the one that is in shortest supply relative to demand. Assuming uptake ratios of the different nutrients don't vary dramatically between regions, basically the authors state Fe is not high in the SO because there never was much to begin with, whereas the other nutrients are high because they are abundantly supplied. Stating the exchange of surface and deep water is prevented in the Arctic is inaccurate, it is an important region of deep water formation.

References (not cited in ms)

Achterberg, E.P., Steigenberger, S., Marsay, C.M., LeMoigne, F.A.C., Painter, S.C., Baker, A.R., Connelly, D.P., Moore, C.M., Tagliabue, A. and Tanhua, T., 2018. Iron Biogeochemistry in the High Latitude North Atlantic Ocean. Scientific Reports, 8(1): 1283. Bridgestock, L., van de Flierdt, T., Rehkämper, M., Paul, M., Middag, R., Milne, A., Lohan, M.C., Baker, A.R., Chance, R., Khondoker, R., Strekopytov, S., Humphreys-Williams, E., Achterberg, E.P., Rijkenberg, M.J.A., Gerringa, L.J.A. and de Baar, H.J.W., 2016. Return of naturally sourced Pb to Atlantic surface waters. Nature Communications, 7: 12921. Middag, R., de Baar, H.J.W., Laan, P., Cai, P.H. and van Ooijen, J.C., 2011. Dissolved manganese in the Atlantic sector of the Southern Ocean. Deep-Sea Research Part Ii-Topical Studies in Oceanography, 58(25-26): 2661-2677. Middag, R., van Hulten, M.M.P., Van Aken, H.M., Rijkenberg, M.J.A., Gerringa, L.J.A., Laan, P. and de Baar, H.J.W., 2015. Dissolved aluminium in the ocean conveyor of the West Atlantic Ocean: Effects of the biological cycle, scavenging, sediment resuspension and hydrography. Marine Chemistry, 177, Part 1: 69-86. Rijkenberg, M.J.A., Middag, R., Laan, P., Gerringa, L.J.A., van Aken, H.M., Schoemann, V., de Jong, J.T.M. and de Baar, H.J.W., 2014. The Distribution of Dissolved Iron in the West Atlantic Ocean. Plos One, 9(6). Roshan, S., DeVries, T., Wu, J. and Chen, G., 2018. The Internal Cycling of Zinc in the Ocean. Global Biogeochemical Cycles, 32(12): 1833-1849. Roshan, S. and Wu, J., 2015a. Cadmium regeneration within the North Atlantic. Global Biogeochemical Cycles, 29(12): 2082-2094. Roshan, S. and Wu, J., 2015b. Water mass mixing: The dominant control on the zinc distribution in the North Atlantic Ocean. Global Biogeochemical Cycles, 29(7): 1060-1074. Weber, T., John, S., Tagliabue, A. and DeVries, T., 2018. Biological uptake and reversible scavenging of zinc in the global ocean. Science, 361(6397): 72-76.

---

## Referee Comment (RC2) · Anonymous Referee #2 · 24 Nov 2020

It is very timely to begin to analyse and synthesize interpretations of the emerging data sets from the GEOTRACES program. This study uses GEOTRACE data from the Atlantic basin to address the hypothesis that "All bioutilised elements are present at higher concentration in high latitude than in low latitude surface waters".

Discussions of elemental vertical profiles have often noted "nutrient-like" profiles for some elements, the concentrations of which increase with depth (e.g.Nozaki, 2001; Froelich, 2014; numerous subsequent references to the "periodic table of elemental profiles". Since the ocean's vertical density stucture is largely controlled by temperature, so denser (deeper) water-masses outcrop at colder temperatures (higher latitudes) bringing with them higher concentrations of nutrient-like tracers mixed along isopycnals. So the basic hypothesis is uncontroversial. I found it odd that this simple mechanistic interpretation wasn't discussed until the very end of the manuscript however. The focus on latitude as the correlate was, for me, a concern. Latitude is a proxy for a more physically relevant quantity (e.g. temperature, density, buoyancy loss, atmospheric deposition,...).

Reading through the manuscript in its current form I kept asking myself why would I care about this hypothesis unless the mechanistic interpretation is front and centre? Why not try and seek relatinoships with the actual drivers and mechanisms of deep water formation/ventilation or atmospheric deposition, for example? In my view focusing the study on the relationship to latitude misses the chance to make a more mechanistic and physical interpretation and could lead uninitiated readers into a misleading view of broader oceanographic understanding.

The contrast between Fe, Al and the other nutrient-like elements is stark and interesting to see in this context. Presumably the role of atmospheric sources at lower latitudes is important (?) but (I felt surprisingly) this is not discussed in Section 4.4 ("Input of deep water is not the only process").

Detailed Comments:

Introduction: Lines 15-25.

What about temperature, salinity, density as contextual discussion? How does surface density vary with latitude? This seems intimately tied to the hypothesis posed here.

Line 20: Citing Levitus et al (1993) for the major basin scale pattern of surface nutrients is not wrong, but surely these patterns were known long before the 1990's and a primary source from early nutrient surveys could be cited (and would be interesting).

Line 25: The discussion about the accuracy of phosphate measurements seems spurious. The major basin-scale gradient moves from $\sim 0$ to $> 1$ or 2 micromolar. The

latitudinal pattern which is the focus of this discussion doesnt depend on the difference between nano- and pico-molar accuracy.

Line 30: The citation to Moore (2016) again seems relevant but not primary. The understanding that upwelling waters are depleted in Fe relative to macronutrients (and why) can be attributed to earlier, or primary sources. (Martin? Archer and Johnson?)

Line 35-40: Again timescale and citations: "the last 20 years" - what about GEOSECS, now 40 years ago? Probably should cite Key et al (2004) for the original GLODAP paper.

Line 55-60: Follows and Williams (2011) should be Williams and Follows (2011).

Line 65: The introduction of a one sentence discussion of Nickel at this point seems very random and odd. What about the data of Nozaki (2001) from the Pacific, which provides a great deal of information about a number of elements? Why isnt it brought into this discussion/introduction?

Line 75-80: Would be useful to quote Broecker and Peng's definitions since you are discussing them.

Line 80-85:a It seems that you are classifying elements that are known to be biologically essential as "unutilized" because their vertical profile is not nutrient-like (i.e. doesnt increase with depth). Why use the word "utilized" at all? Why not define a different category for your hypothesis-test (e.g. "nutrient-like", i.e. increasing concentration with depth) because that's what you are actually examining. As it stands, it appears that you are redifining the meaning of "bioutilized" to to support your hypothesis - because its almost certain that tracers that increase with depth will have higher concentrations at surface high-latitude outcrops. Overall, it seems to me that the classification scheme employed here is misleading and needs to be changed.

Summary:

This study addresses newly available, large scale tracer data sets and is thus timely.

However, the focus on latitude-tracer correlations, without clear reference to the underlying mechanistic links, did not provide a useful concept or tool for me. The oceanographic context that appears as the discussion should have, in my opinion, been part of the introduction and motivation for seeking relationships with more meaningful correlates. Citation of relevant earlier work could be more thorough. The contrast between Fe, Al and other nutrient like tracers is brought forth clearly in the data analysis, but the underlying reasons are not really discussed. Overall, in my opinion, this manuscript and study, in its current form, represents a good start but needs quite a bit of work to reach its potential value.

---

## Author Comment (AC1) · 8 Jan 2021

**Response to Anonymous Referee #2**

We thank referee #2 for the helpful suggestions in their review. We address point by point the concerns of the referee.

**General Comments:**

*It is very timely to begin to analyse and synthesize interpretations of the emerging data sets from the GEOTRACES program. This study uses GEOTRACE data from the Atlantic basin to address the hypothesis that "All bioutilised elements are present at higher concentration in high latitude than in low latitude surface waters".*

*Discussions of elemental vertical profiles have often noted "nutrient-like" profiles for some elements, the concentrations of which increase with depth (e.g. Nozaki, 2001; Froelich, 2014; numerous subsequent references to the "periodic table of elemental profiles". Since the ocean's vertical density stucture is largely controlled by temperature, so denser (deeper) water-masses outcrop at colder temperatures (higher latitudes) bringing with them higher concentrations of nutrient-like tracers mixed along isopycnals. So the basic hypothesis is uncontroversial. I found it odd that this simple mechanistic interpretation wasn't discussed until the very end of the manuscript however. The focus on latitude as the correlate was, for me, a concern. Latitude is a proxy for a more physically relevant quantity (e.g. temperature, density, buoyancy loss, atmospheric deposition,...). Reading through the manuscript in its current form I kept asking myself why would I care about this hypothesis unless the mechanistic interpretation is front and centre? Why not try and seek relationships with the actual drivers and mechanisms of deep water formation/ventilation or atmospheric deposition, for example? In my view focusing the study on the relationship to latitude misses the chance to make a more mechanistic and physical interpretation and could lead uninitiated readers into a misleading view of broader oceanographic understanding.*

We will cite Nozaki's 2001 seminal work where we talk about classification of elements into different categories but note that Nozaki's primary focus was vertical distributions whereas ours is horizontal distributions. We agree that the basic hypothesis is uncontroversial to some degree but it has not previously been subjected to statistical testing or to a comprehensive examination across such a range of elements, because this has only now started to become possible thanks to the emerging datasets from the GLODAP and most recently the GEOTRACES syntheses. We agree that the mechanisms behind the latitude-concentration relationship need to be discussed earlier. We thank the referee for their suggestion to examine relationships with physical parameters such as temperature and density and agree that doing so will considerably enhance the interest and usefulness of our study.

*The contrast between Fe, Al and the other nutrient-like elements is stark and interesting to see in this context. Presumably the role of atmospheric sources at lower latitudes is important (?) but (I felt surprisingly) this is not discussed in Section 4.4 ("Input of deep water is not the only process").*

We are obviously aware of this process but omitted to include it.

**Specific Comments:**

*Introduction: Lines 15-25. What about temperature, salinity, density as contextual discussion? How does surface density vary with latitude? This seems intimately tied to the hypothesis posed here.*

We agree to add correlations to these physical parameters in order to provide insights into the drivers of the correlations between latitude and elemental concentrations.

*Line 20: Citing Levitus et al (1993) for the major basin scale pattern of surface nutrients is not wrong, but surely these patterns were known long before the 1990's and a primary source from early nutrient surveys could be cited (and would be interesting).*

As noted in our response to reviewer #1, we make a distinction between patterns that are suspected on the basis of limited data, and patterns that are understood on the basis of datasets of a sufficient size to define the global distribution. However, we accept that we could have done a better job in citing previous studies, for instance papers arising from earlier surveys such as GEOSECS and TTO.

*Line 25: The discussion about the accuracy of phosphate measurements seems spurious. The major basin-scale gradient moves from ~0 to > 1 or 2 micromolar. The latitudinal pattern which is the focus of this discussion doesnt depend on the difference between nano- and pico-molar accuracy.*

We accept this point.

*Line 30: The citation to Moore (2016) again seems relevant but not primary. The understanding that upwelling waters are depleted in Fe relative to macronutrients (and why) can be attributed to earlier, or primary sources. (Martin? Archer and Johnson?)*

We see the reviewer's point and should have cited some earlier work but we emphasise again the difference between a compelling argument based on analysis of a comprehensive dataset, in contrast to a plausible suggestion based on limited data. Earlier investigators simply did not have access to the large datasets that have recently become available.

*Line 35-40: Again timescale and citations: "the last 20 years" - what about GEOSECS, now 40 years ago? Probably should cite Key et al (2004) for the original GLODAP paper.*

Yes.

*Line 55-60: Follows and Williams (2011) should be Williams and Follows (2011).*

Yes.

*Line 65: The introduction of a one sentence discussion of Nickel at this point seems very random and odd. What about the data of Nozaki (2001) from the Pacific, which provides a great deal of information about a number of elements? Why isnt it brought into this discussion/introduction?*

The point is to acknowledge the recent work on nickel, especially because measured along an Atlantic-long transect. As mentioned above, we will cite Nozaki's 2001 seminal work where we talk about classification of elements into different categories but note that Nozaki's primary focus was vertical distributions whereas ours is horizontal distributions.

*Line 75-80: Would be useful to quote Broecker and Peng's definitions since you are discussing them.*

This could be done.

*Line 80-85:a It seems that you are classifying elements that are known to be biologically essential as "unutilized" because their vertical profile is not nutrient-like (i.e. doesnt increase with depth). Why*

*use the word "utilized" at all? Why not define a different category for your hypothesis-test (e.g. "nutrient-like", i.e. increasing concentration with depth) because that's what you are actually examining. As it stands, it appears that you are redifining the meaning of "bioutilized" to to support your hypothesis - because its almost certain that tracers that increase with depth will have higher concentrations at surface high-latitude outcrops. Overall, it seems to me that the classification scheme employed here is misleading and needs to be changed*

As mentioned in the response to reviewer #1, we agree that bio(un)utilised is slightly clumsy and will replace with nutrient-like for those elements we had previously stated as bioutilised and non-nutrient-like for what we had called biounutilised. This is better but also imperfect (iron is a nutrient). Most important is that we define our use of the term clearly and unambiguously. We already do this.

References (not stated in MS)

Key, R. M., Kozyr, A., Sabine, C. L., Lee, K., Wanninkhof, R., Bullister, J. L., Feely, R. A., Millero, F. J., Mordy, C. and Peng, T.-H.: A global ocean carbon climatology: Results from Global Data Analysis Project (GLODAP), Global Biogeochem. Cy. 18, doi:10.1029/2004GB002247, 2004.

Nozaki, Y.: 2001. Elemental Distribution in Encyclopedia of Ocean Sciences, edited by: Steele, J. H., Academic Press, 840-845, 2001.

---

## Author Comment (AC2) · 8 Jan 2021

**Response to Anonymous Referee #1**

We thank referee #2 for the helpful suggestions in their review. We address point by point the concerns of the referee.

**General Comments:**

*This manuscript about latitudinal patterns of trace elements is generally well written. It is based on existing data from the GEOTRACES program and aims to test the hypothesis that nutrient type elements occur at higher concentrations at higher latitude, notably the Southern Ocean. The fact that they were able to proof this hypothesis is not at all surprising to me given that nutrient type elements are also referred to as 'accumulated' type elements as they accumulate in older (deep) water. Besides the nutrient type profile (low in surface waters and concentrations that increase with increasing depth) this also leads to a well-known and strong interbasin fractionation where concentrations are higher in the old deep North Pacific or deep Southern Ocean compared to the relatively young deep North Atlantic. As acknowledged in the introduction and discussion of this ms, upwelling of old deep water in the Southern Ocean thus leads to supply of macronutrients. However, this inherently also supplies other nutrient type (trace) elements to surface waters (but not Fe that is subject to scavenging, hence has a hybrid type distribution (Bruland et al., 2014)), and Fe limitation results in 'left-over' nutrients. In the North Atlantic, deep mixing also leads to supply of nutrient type elements to surface waters, albeit lower than compared to the Southern Ocean due to lower deep water concentration in the Atlantic, and seasonal Fe limitation (e.g. Achterberg et al., 2018) results in some 'left-over' nutrients. So while the authors did prove their hypothesis using statistical tests, this hypothesis is actually a well-established concept, not only for the macro nutrients, but also the 'nutrient-type' trace metals (hence their classification as nutrient-type aka as recycled or accumulated type). As far as I can tell, the conclusions of this manuscript are also a main message of any chemical oceanography text book, except for the lines on the Arctic where the authors seemingly missed that the position in the global conveyor (with related absence of old deep water that is strongly enriched in nutrient type elements) is important. Moreover, established concepts regarding the importance of sources, sinks and chemistry of different elements are ignored and I disagree with the notion that recent work did not focus on latitudinal patterns (see specific comments).*

We accept that we could have more comprehensive in citing previous work (there is always a balance between being overly concise vs citing too much of the literature but we agree that we did not find the right balance). However, that said, we do not agree at all that our conclusions can be found in any chemical oceanography textbook. Likewise, we did not miss the importance of position on the deep conveyor as it was mentioned in the paper (lines 343-344 and 396). Furthermore, for those parts of the Arctic (particularly on the western, i.e. Pacific side) where there is a strong halocline that is more or less impossible to break down, the position on the deep conveyor is irrelevant because surface and deep waters do not mix to any great extent. While we thank the reviewer for the thorough comments and extra references, and accept some of the comments, on the whole we do not feel that this is a balanced and fair evaluation of our manuscript.

*Overall, I'm afraid I do not see any novel contribution of this manuscript and therefore cannot recommend it for publication in its current form.*

Our paper is clearly a novel contribution and we do not think it is reasonable to suggest otherwise. While we agree that there is previous work along these lines, in particular for macronutrients, and that

we did not acknowledge it as best we could, nevertheless there are 2 aspects of our paper that are without doubt novel:

(1) the extension of the concept of high-latitude enrichment from macro-nutrients and some trace elements to a general rule applying to all macro-nutrients, all "bio-utilised" trace metals and also to DIC and TA. High-latitude enrichment for nDIC and nTA has only been demonstrated recently, in the work of our (Prof Tyrrell's) group (Wu et al., 2019; Fry et al., 2015). This advance has only been possible since the advent of the GLODAP database and was not appreciated for DIC and TA earlier. Only now, with GEOTRACES, is it becoming possible to extend findings from a few trace elements to a wider picture.

(2) As indeed the reviewer acknowledges, our statistical analysis advances this claim of high-latitude enrichment from a verbal one (a so-called "armchair suggestion") to a statistically proven one. While earlier papers may have proposed the hypothesis, we demonstrate it to be true with very high confidence levels (p < 0.001). This is an important step and by itself contradicts the reviewer's claim of no novel contribution.

**Specific comments:**

*Line 16 distributions of elements in the oceans (there are many distributions that were understood much earlier)*

We make a distinction between distributions that were _suspected_, for instance where there was some data _suggesting_ an overall pattern, and between those that were _understood_. For the latter, a dataset of significant size, sufficient to define the global distribution, is required. We disagree that such datasets were available for macronutrients before the dates mentioned, and therefore that the distributions were understood before those dates.

*Intro Jumps straight into macro nutrient distributions followed by alkalinity without any context or connection between the subsections*

A "sign-posting" sentence would indeed be helpful.

*Line 29 iron and light limited*

Yes, in the sense of a proximate limiting factor, light is often also important because of the deep mixed layer depths in the Southern Ocean. Nevertheless, iron is observed to be drawn down to limiting levels, and therefore it is lack of iron, not lack of light, that sets the limit to the amount of phytoplankton production over the course of the spring/summer growing season.

*Line 52 awkward sentence.*

We will rephrase.

*Line 64/65 what is the point of this standalone sentence? Similar observations for Cd and Zn by the way*

The point is to acknowledge the recent work on nickel, especially because measured along an Atlantic-long transect. We agree that we should also acknowledge Middag's work on cadmium and zinc.

*Line 69-70 for Ni is was attributed to upwelling of deep water (direct citation: 'The higher concentrations in the Southern Ocean are most likely due to upwelling of older deep water in this region whereas in contrast, the Arctic is largely supplied by nutrient poor surface water transported north with the Gulf stream' and also depicted in figure 7 of this paper. Similar arguments for Cd and Zn in Middag et al., 2019, 2020)*

This again relates to the distinction between distributions that are well-known because of large datasets, and distributions that are suspected based on smaller amounts of data. These are two very different "levels of knowledge". The main point of this paper is the emerging general rule that all bio-utilised elements show high latitude elevation of values. It is therefore important to be precise about which elements are already shown to exhibit this latitudinal pattern and those which are only suspected to exhibit it.

In terms of the point about the Arctic being supplied by surface water via the Gulf Stream, we note that this is only true for one side of the Arctic, the Atlantic or eastern side. The western (Pacific) side is supplied by surface water coming through the Bering Straits.

*Line 75-80 I find it extremely odd to call the bio-essential element Mn 'biounutilised' whereas it has been shown to limit productivity in the Southern Ocean. Actually, in the Southern Ocean, Mn would be classified as bioutilised (probably bio-utilised is more readable) as concentrations are depleted in the surface and increase with depth (e.g. Middag et al., 2011), whereas Fe in parts of the equatorial Atlantic would be biounutilised (probably bio-unutilised is more readable) as concentrations are elevated in the surface and decrease with depth (e.g. Rijkenberg et al., 2014).*

We agree that it is slightly clumsy and will replace with "nutrient-like". This is better but also imperfect (iron is a nutrient). Most important is that we define our use of the term and do so unambiguously. We already do this.

*Line 89 to what salinity is the data normalized?*

35.

*Line 110 table 1 why is Mn data from GA02 and GIPY 05 ignored?*

Mn was inadvertently omitted from the Table.

*Line 190 why was this based on one individual station (see also previous comment)*

This follows previous practice (e.g. Broecker & Peng 1982). The South Atlantic is fairly typical in many regards, for instance not iron-limited and not subject to unusually large dust inputs like the North Atlantic. We have checked whether those elements categorised as bio-utilised shows surface depletion in the majority of all of the GEOTRACES stations (see fig.1), and likewise for those categorised as bio-unutilised. Our analysis finds the list to be unaltered when calculated this way.

[Figure]

*Figure 1 - Depth profiles of each element. Blue data represents all of the processed data for each element used within the study, with red data points representing every 20th data point. The prefix 'n' indicates salinity normalisation of concentrations.*

*Line 310 this distribution of Mn and Fe is well known and related to the chemistry of the elements (both subject to oxidative scavenging), biological utilization and notably the presence of strong sources at low latitude (mainly Saharan dust deposition at low latitude, but also fluvial input and reducing sediments) whereas these sources are lacking or much reduced at higher latitudes. The fact that Mn and Fe are low in the HNLC Southern Ocean is something that can be found in any text book or review paper on chemical oceanography addressing these elements.*

This sentence is not only about the Southern Ocean but rather about high latitudes in general.

*Line 312 For Al and Pb this distribution (e.g. Bridgestock et al., 2016; Middag et al., 2015) is well known and again related to sources and sinks in its biogeochemical cycling.*

We agree, but do not understand the reviewer's point with respect to line 312. Furthermore, if statements in the paper are in complete agreement with textbooks than we do not understand the need for this to be commented on, since they cannot be contentious. Obviously, if the main conclusions of the paper are already included in textbooks then there is no novelty in the paper, but this is not the situation here (see above).

*Section 4.3. This is basically a brief summary of a text book on chemical oceanography (As a matter of fact, one of the re-occurring questions I ask in the exam about my chemical oceanography lectures is to explain the higher concentrations of nutrient type elements in the higher latitude regions, notably the Southern Ocean)*

We agree that this is well understood for the Southern Ocean but disagree that it is so well understood as a general rule for the global ocean, in particular for the North Pacific where the physical mechanisms that bring about exchange of surface water with deep water are considerably more complex. We accept that the phrasing of this section could be improved to make it clearer which aspects are well-known and which not so much.

*Line 355 excess of evaporation over precipitation should be accounted for in the salinity normalization.*

It is, as noted in the example of total alkalinity before salinity normalisation.

*Line 356-357 similar for Mn; and presence and absence of sources such as atmospheric dust, (reducing) sediment, fluvial input, anthropogenic sources etc.*

Yes.

*4.5 Deep waters in Arctic Ocean are also not particularly enriched in nutrient type elements like the Southern Ocean as deep waters here are much younger, i.e. Arctic Ocean sits mainly at the beginning of the ocean conveyor with inflow of nutrient poor Atlantic surface water and only modest amounts of old pacific deep water (see large body of GEOTRACES work in Arctic from both during IPY as well as recent expeditions)*

Again, this is only true for the Atlantic side of the Arctic. The start of the deepwater conveyor belt is in the Nordic (Greenland, Icelandic and Norwegian) Seas. Deep water is not formed in the Pacific (western) side of the Arctic. Given that we state that we are referring in this section to the western side of the Arctic where there is a strong halocline, the composition of the deep waters is not relevant.

*Line 375-3-77 a main point of those recent papers was the importance of high nutrient (incl nutrient type trace metals) high latitude waters and their influence on both the horizontal (meridional) and vertical distributions (and coupling between elements) at lower latitudes.*

*Line 389 this was a main conclusion of many recent papers (e.g. Middag et al., 2019; Middag et al., 2020; Middag et al., 2018; Roshan et al., 2018; Roshan and Wu, 2015a; Roshan and Wu, 2015b; Vance et al., 2017; Weber et al., 2018) and the lack of Fe supply relative to macro nutrients in upwelling regions is about as old as the term 'HNLC'.*

We will add acknowledgement of this previous work. At the same time we reiterate the point made at the beginning of our response: while this rule has been proposed previously, it has not been

*demonstrated statistically* before, as we have done here. Additionally, it has not been established as a general rule including DIC and TA as well as numerous trace elements.

*4.7 point 2; given the absence of a strong dust source over large parts of the Pacific, there will be differences for some elements (e.g. the high concentrations of Al, Fe and Mn at low latitude are not found). Moreover, part of the equatorial Pacific is an HNLC region with elevated concentrations of nutrient type elements*

Yes.

*point 3; this is well known, hence the high-latitude North Pacific is a HNLC region whereas the high latitude North Atlantic only has minor inventories of 'left-over' macro nutrients at the end of the phytoplankton growth season and only experiences seasonal Fe limitation (end of season).*

It is well-known for macronutrients but not for other bio-utilised elements.

*point 4: except those with a strong fluvial influence, see recent work on metals in the Arctic trans polar drift. Also noted in recent work on global or Atlantic distribution of Cd, Zn an Ni.*

We make the prediction because it follows logically from the theme of this paper. We nevertheless acknowledge (section 4.4) that other processes can intervene.

*Conclusions The statement 'presumably because of its role as the limiting nutrient for primary production in upwelling regions' does not explain anything; the limiting nutrient is the one that is in shortest supply relative to demand. Assuming uptake ratios of the different nutrients don't vary dramatically between regions, basically the authors state Fe is not high in the SO because there never was much to begin with, whereas the other nutrients are high because they are abundantly supplied. Stating the exchange of surface and deep water is prevented in the Arctic is inaccurate, it is an important region of deep water formation.*

We disagree: assuming primary production continues until one nutrient or another runs out, and the nutrient that runs out first is iron in the high-latitude North Pacific and the Southern Ocean, then that can indeed explain why iron does not in general follow the rule of high latitude enrichment. Exhaustion of the proximate limiting nutrient prevents it. We will change the last sentence to read "western Arctic halocline".

References (not cited in MS)

Achterberg, E.P., Steigenberger, S., Marsay, C.M., LeMoigne, F.A.C., Painter, S.C., Baker, A.R., Connelly, D.P., Moore, C.M., Tagliabue, A. and Tanhua, T.: Iron Biogeochemistry in the High Latitude North Atlantic Ocean. Sci. Rep-UK, 8, doi:10.1038/s41598-018-19472-1, 2018

Bridgestock, L., van de Flierdt, T., Rehkämper, M., Paul, M., Middag, R., Milne, A., Lohan, M.C., Baker, A.R., Chance, R., Khondoker, R., Strekopytov, S., Humphreys-Williams, E., Achterberg, E.P., Rijkenberg, M.J.A., Gerringa, L.J.A. and de Baar, H.J.W.: Return of naturally sourced Pb to Atlantic surface waters. Nat. Commun., 7, doi:10.1038/ncomms12921, 2016

Middag, R., de Baar, H.J.W., Laan, P., Cai, P.H. and van Ooijen, J.C.: Dissolved manganese in the Atlantic sector of the Southern Ocean. Deep-Sea Res., Pt. II, 58, 2661-2677, doi:10.1016/j.dsr2.2010.10.043, 2011.

Middag, R., van Hulten, M.M.P., Van Aken, H.M., Rijkenberg, M.J.A., Gerringa, L.J.A., Laan, P. and de Baar, H.J.W.: Dissolved aluminium in the ocean conveyor of the West Atlantic Ocean: Effects of the biological cycle, scavenging, sediment resuspension and hydrography. Mar. Chem., 177, 69-86, https://doi.org/10.1016/j.marchem.2015.02.015, 2015.

Rijkenberg, M.J.A., Middag, R., Laan, P., Gerringa, L.J.A., van Aken, H.M., Schoemann, V., de Jong, J.T.M. and de Baar, H.J.W.: The Distribution of Dissolved Iron in the West Atlantic Ocean. Plos One, 9, doi:10.1371/journal.pone.0101323, 2014.

Roshan, S., DeVries, T., Wu, J. and Chen, G.: The Internal Cycling of Zinc in the Ocean. Global Biogeochem. Cy., 32, 1833-1849, doi:10.1029/2018GB006045, 2018.

Roshan, S. and Wu, J.: Cadmium regeneration within the North Atlantic. Global Biogeochem. Cy., 29, 2082-2094, doi:10.1002/2015GB005215, 2015a.

Roshan, S. and Wu, J.: Water mass mixing: The dominant control on the zinc distribution in the North Atlantic Ocean. Global Biogeochem, Cy., 29, 1060-1074, doi:10.1002/2014GB005026, 2015b.

Weber, T., John, S., Tagliabue, A. and DeVries, T.: Biological uptake and reversible scavenging of zinc in the global ocean. Science, 361, doi:10.1126/science.aap8532, 72-76, 2018.